# Home-based screening tools for amblyopia: a systematic review protocol

Samantha Sii ![ORCID],[1] Chung Shen Chean,[2] Helen J Kuht,[3] Mervyn G Thomas,[3] Sohaib R Rufai ![ORCID] [3,4]

MGT and SRR are joint senior authors.

¹Department of Ophthalmology, Kettering General Hospital NHS Trust, Kettering, UK
²Department of Ophthalmology, Northampton General Hospital, Northampton, UK
³University of Leicester Ulverscroft Eye Unit, Leicester Royal Infirmary, Leicester, UK
⁴Clinical and Academic Department of Ophthalmology, Great Ormond Street Hospital for Children, London, UK

**Correspondence to**
Dr Sohaib R Rufai;
Sohaib.Rufai@nhs.net and
Dr Mervyn G Thomas;
mt350@leicester.ac.uk

## ABSTRACT

**Introduction** Amblyopia is an important public health concern associated with functional vision loss and detrimental impact on the physical and mental well-being of children. The gold standard for diagnosis of amblyogenic conditions currently involves screening by orthoptists and/or ophthalmologists. The bloom of technology enables the use of home-based screening tools to detect these conditions at an early stage by the layperson in community, which could reduce the burden of screening in the community, especially during restrictions associated with the COVID-19 pandemic. Here, we propose a systematic review aiming to evaluate the accuracy and reliability of home-based screening tools compared with the existing gold standard.

**Methods and analysis** We aim to search for studies involving home-based screening tools for amblyopia among children aged under 18 years. Oxford Centre for Evidence-Based Medicine Level 4 evidence and above will be included, without language or time restrictions. The following platforms will be searched from inception to 31 August 2021: PubMed, Medline, The Cochrane Library, Embase, Web of Science Core Collection and Clinicaltrials.gov. Two independent reviewers will identify studies for inclusion based on a screening questionnaire. The search and screening will start on 14 August 2021 until 1 October 2021. We aim to complete our data analysis by 30 November 2021. Risk of bias will be assessed using the Quality Assessment of Diagnostic Accuracy Studies 2 (QUADAS-2) tool for diagnostic accuracy studies only. Our primary outcome measure is the diagnostic accuracy of home-based screening tools, while secondary outcome measures include validity, feasibility, reproducibility and cost-effectiveness, where available.

**Ethics and dissemination** Ethical approval is not necessary as no primary data will be collected. The findings will be disseminated through presentations at scientific meetings and peer-reviewed journal publication.

**PROSPERO registration number** CRD42021233511.

## Strengths and limitations of this study

► This will be the first systematic review evaluating the accuracy and reliability of home-based screening tools for amblyopia.
► Published and unpublished literature without language or time restrictions will be included.
► Protocol methodology is based on principles extracted from the Cochrane Collaboration.
► The main limitation could be a scarcity of randomised controlled trials and diagnostic accuracy studies involving home-based screening tools.
► The broad search strategy should help ensure that all relevant literature is included.

## INTRODUCTION

Amblyopia is one of the most common preventable causes of vision loss affecting children. It continues to represent a significant public health concern, affecting 2–5% of the population.[1–3] Amblyopia is usually associated with visual deprivation early in life,[4] due to amblyogenic risk factors which include uncorrected refractive errors, astigmatism, congenital pathologies or media opacities that causes stimulus deprivation, and abnormal binocular interaction from strabismus.[5–7] Children with amblyopia are characterised by monocular or binocular visual deficits, including reduced visual acuity, contrast sensitivity, contour integration and depth perception without observable ocular pathological features.[8]

Amblyopia is largely asymptomatic initially, but untreated amblyopia resulting in vision loss can lead to problems at school, bullying, reduced quality of life, lifelong consequences on future occupation choices and mental health issues.[9 10] Contrary to the traditional notion that amblyopia treatment may be ineffective for children above 7 years old,[11] the Paediatric Eye Disease Investigator Group studies showed that treatment of amblyopia may still be effective in children aged 7–17 years,[12 13] with the effectiveness of treatment becoming significantly reduced with time.[14] While amblyopia is treatable, the key to manage this disorder effectively is early detection by screening. Screening for amblyopia was introduced in the 1950s and advocated in many countries.[15] Many screening programmes have been unsuccessful, with an estimation of less than 25% of preschool-aged children being screened through a government or private programme in the USA.[16]

In addition, up to 60% of primary care providers do not perform vision screening on preschool-aged children, and others perform screening inconsistently.[16] Significant barriers to traditional vision screening include cost, limited access to healthcare and a limited number of qualified screeners available.[17] Hence, a variety of methodologies for vision screening have been trialled, including the use of home-based amblyopia screening tools, to help overcome these barriers to vision screening.[18]

The COVID-19 pandemic illustrates the increasingly important role of telemedicine as a method of clinician–patient interaction. The use of home-based screening tools for amblyopia are increasingly advocated as social distancing is practised to minimise the risk of viral transmission.[19 20] Furthermore, COVID-19 related restrictions and lockdowns may have resulted in many children missing out opportunities for amblyopia screening.[19] Home-based screening may offer a solution,[21] but this has not been rigorously assessed and evaluated by systematic review. Here, we propose a systematic review to evaluate home-based amblyopia screening tools.

## METHODS AND ANALYSIS
This protocol is drafted according to the Preferred Reporting Items for Systematic Review and Meta-Analysis Protocols (PRISMA-P) checklist.[22]

### Eligibility criteria for studies
The eligible study characteristics for this systematic review are defined according to the Population, Intervention, Comparison, Outcome and Study Design[23] study strategy outlined in table 1.

### Information sources
The following electronic searches will be included in this systematic review:

1. Ovid MEDLINE 1946 to present.
2. PubMed.
3. The Cochrane Library.
4. Embase 1974 to present.
5. Web of Science Core Collection (1970 to present).
6. Clinicaltrials.gov.

Sources 1 and 4 will be searched through the Ovid platform separately.

### Other sources
Publications of all formats, including protocols and conference abstracts, not limited by year and language will be included.

To ensure literature saturation, references of included studies will be searched and included if meeting inclusion criteria. Authors of studies with insufficient data published will be contacted via email in attempt to extract relevant outcome data. If there is no response from these authors after 14 days, another email will be sent to attempt to establish contact. If there is still no response after 14 days, these studies will be excluded.

### Search strategy
The search strategy was developed after convening with a research services consultant with experience in systematic review. The search terms 'amblyopia', 'visual acuity', 'vision screening', 'home', 'web', 'internet' 'app', 'smartphone' and 'mobile' were entered into the electronic search platforms. A sample of the full search strategy using the electronic databases listed is available in online supplemental appendix 1.

### Study records
#### Data management
EndNote V.X9 (Thomson Reuters, New York, New York, USA) reference management software will be used for data management.

| Table 1 | Eligibility criteria | |
|---|---|---|
| **PICOS strategy** | **Inclusion criteria** | **Exclusion criteria** |
| Population | Studies involving screening for amblyopia in children aged under 18 years old | Studies involving adults aged 18 years old and above |
| Intervention | Home-based screening tools including: (1) internet or web-based visual acuity screening tools; (2) mobile applications used to screen for conditions contributing to amblyopia; (3) novel home-based gadgets or instruments used to screen for conditions contributing to amblyopia | Orthoptist-led or ophthalmologist-led amblyopia screening tests including: (1) standard logMAR (or equivalent) visual acuity measurement charts; (2) comprehensive eye examination using slit lamp or ocular motility examination; (3) autorefractors or photoscreeners |
| Comparison/control | Orthoptist-led or ophthalmologist-led amblyopia screening | Not applicable |
| Outcomes | Primary outcome measure: diagnostic accuracy of home-based amblyopia screening tools. Secondary outcome measures, where available: validity, feasibility, reproducibility, cost-effectiveness | (1) Studies not reporting outcomes related to amblyopia screening; (2) epidemiological studies reporting prevalence of amblyopia |
| Study design | According to the Oxford Centre for Evidence-Based Medicine (CEBM) Level 4 evidence and above will be included[26] | CEBM Level 5 evidence and below will be excluded |

PICOS, Population, Intervention, Comparison, Outcome and Study Design.

## Selection of studies

Two independent screeners (SS and CSC) shall follow a three-stage screening method, according to a screening questionnaire (online supplemental appendix 2). After the screening of titles and abstracts, SS and CSC will compare and attempt to resolve any disagreements on the inclusion of articles, where applicable. If any disagreement remains, opinion will be sought from the third arbitrator (HK). We aim to start the search by 14 August 2021 and complete the screening process by 1 October 2021.

## Data collection

Our data collection tool adapted from the Cochrane Collaboration is included in online supplemental appendix 3. A preliminary data collection form was first drawn and piloted among the authors of this study before use. The following data shall be collected: study design, number of included patients, duration of study, method of intervention used, index test and reference test where applicable. Outcomes pertinent to the quality of diagnostic studies including investigators conducting test, subjects receiving test, method of interpretation of test, blinding of participants or investigators and withdrawal rate will also be included.

## Outcome measures and prioritisation

Our primary outcome measure of interest will be the diagnostic accuracy of home-based screening tools in detecting amblyopia compared with the existing gold standard which is diagnosis made by orthoptists or ophthalmologists. Outcomes from diagnostic accuracy studies such as sensitivity, specificity, positive predictive value (PPV) and negative predictive value (NPV) will be prioritised as the primary outcome as they will translate into meaningful endpoints for comparing the effectiveness of home-based screening tools against the gold standard. The secondary outcome measures, or surrogate measures of this review where available may include validity, feasibility, reproducibility and cost-effectiveness of these home-based screening tools compared with existing gold standard screening. These will be reported in appropriate statistical measures if represented by studies with large enough sample sizes. As some outcomes may be reported as a composite measure, we will extract all composite and individual outcomes as reported in the studies.

## Risk of bias assessment

Risk of bias assessment will be done for diagnostic accuracy studies only. The quality of diagnostic accuracy studies will be assessed using the Quality Assessment of Diagnostic Accuracy Studies 2 (QUADAS-2) tool (online supplemental appendix 4).[24] These judgements will be made independently by two review authors (SS, CSC) and any disagreements discussed with the third arbitrator (HK). If our risk of bias assessment shows lack of good quality studies with adequate sample sizes, statistical measures will not be summarised quantitatively and vice versa.

## Data analysis

Scoping searches suggest that mainly observational studies will be returned by our search strategy with few relevant randomised controlled trials (RCTs). Weighted means for primary outcome measures (such as sensitivity, specificity, PPV, NPV) will only be calculated if multiple RCTs or good quality large scale prospective studies are identified. Otherwise, we shall perform a qualitative review summarising the available evidence of good quality studies in the form of tables to explain the characteristics of and results of the included studies as well as relevant p values. This will be followed by a narrative synthesis of secondary outcome measures such as validity, feasibility, reproducibility or cost-effectiveness of home-based screening tools. We aim to complete our data analysis by 30 November 2021.

## Confidence in cumulative estimate

The quality of evidence for all outcomes will be judged using the Grading of Recommendations Assessment, Development and Evaluation working group methodology[25] and will be judged as high (further research is very unlikely to change our confidence in the estimate of effect), moderate (further research is likely to have an important impact on our confidence in the estimate of effect and may change the estimate), low (further research is very likely to have an important impact on our confidence in the estimate of effect and is likely to change the estimate) or very low (very uncertain about the estimate of effect).

## Patient and public involvement statement

As this systematic review does not involve recruitment of patients for research, patient and public involvement is not applicable.

## Ethics and dissemination

As this systematic review does not involve recruiting patients, independent ethical approval is not required. The findings of this systematic review shall be disseminated through presentations at scientific meetings, as well as peer-reviewed journal publication. Any data generated from this systematic review will be made available from the corresponding author on reasonable request.

## DISCUSSION

To our knowledge, this is the first systematic review aiming to compare home-based screening tools and existing screening services offered through ophthalmologists and orthoptists to diagnose amblyopia. We adhered to the PRISMA-P checklist in drafting this protocol. Through publication of this protocol, we aim to provide transparency in the methodology of our systematic review. This should increase internal validity by preventing publication bias and should help avoid study duplication.

**Contributors** SS: concept, methodology, drafting of protocol, piloting of questionnaires, revision and final approval of manuscript. CSC: drafting of protocol (introduction) and critically reviewing manuscript. HK: critically reviewing manuscript. MT: supervision, critically reviewing protocol and manuscript. SRR: supervision, concept, methodology, peer-review of search strategy, questionnaires, critically reviewing protocol and manuscript.

**Funding** This work is funded by the National Institute for Health Research (NIHR). SRR is funded by a National Institute for Health Research (NIHR) Doctoral Fellowship Award ID: NIHR300155. MT is also supported by the NIHR (CL-2017-11-003) and the Ulverscroft foundation.

**Disclaimer** This paper presents independent research funded by the NIHR. The views expressed are those of the author(s) and not necessarily those of, the Ulverscroft Foundation, the NHS, the NIHR or the Department of Health and Social Care.

**Competing interests** None declared.

**Patient consent for publication** Not required.

**Provenance and peer review** Not commissioned; externally peer reviewed.

**ORCID iDs**
Samantha Sii http://orcid.org/0000-0002-9511-0717
Sohaib R Rufai http://orcid.org/0000-0001-8134-6393

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
