## [Reviewer comments · BMJ Open]

ARTICLE DETAILS

TITLE (PROVISIONAL)	Home-based screening tools for amblyopia: a systematic review protocol
AUTHORS	Sii, Samantha; Chean, Chung Shen; Kuht, Helen; Thomas, Mervyn; Rufai, Sohaib

VERSION 1 – REVIEW

REVIEWER	Raul E. Ruiz-Lozano Sindh Institute of Ophthalmology and Visual Sciences, Ophthalmology
REVIEW RETURNED	22-May-2021

GENERAL COMMENTS	This is a very original and clearly defined protocol. I believe results will provide valuable information for ophthalmologists regarding the utility of home-based screening tools for amblyopia. When presenting results, I strongly encourage authors to include a brief discussion of the cost-effectiveness of amblyopia management as compared to other ophthalmic diseases (Reference: Busbee BG, Brown GC, Brown MM (2003) Cost-effectiveness of ocular interventions. Curr Opin Ophthalmol 14(3):132–138. https://doi.org/10.1097/00055735-200306000-00004)
--

REVIEWER	Patricia Nelson Texas Tech University Health Sciences Center, Surgery - Ophthalmology
REVIEW RETURNED	25-May-2021

GENERAL COMMENTS	Well thought out study design and well constructed appendix resources. Would be a relevant, impactful, and timely study.
--

REVIEWER	Anna O'Connor University of Liverpool
REVIEW RETURNED	12-Jul-2021

GENERAL COMMENTS	The use of home based devices for assessing children is a very important one and highly pertinent in current times. Many children have yet to be screened from reception class in 2019/2020 with more waiting from the current cohort of children. This protocol is timely and appropriate but the focus of just looking at devices to be used in a screening context is very specific. There is a potential role for devices to monitor VA at home as well, not just for detection purposes. During the pandemic there have been a number of papers evaluating the use of VA testing methods at home, some of which the authors have cited. Is there scope for evaluating the test accuracy overall rather than simply for the detection of amblyopia? To determine accuracy of any test at detecting amblyopia it would require the child to have a further assessment to diagnose
--

	amblyopia. Given the limited number of studies, it would be worthwhile considering expanding to include accuracy in general of VA tests performed at home to include measures such as test retest variability. Two minor points - the British and Irish Orthoptic Journal is on PubMed but not Medline, is there scope for including this journal (and other relevant orthoptics journals)? Search strategy – is there a reason that the term visual acuity isn't included?
--	---

VERSION 1 – AUTHOR RESPONSE

Reviewer: 1

Dr. Raul E. Ruiz-Lozano, Sindh Institute of Ophthalmology and Visual Sciences

Comments to the Author:

This is a very original and clearly defined protocol. I believe results will provide valuable information for ophthalmologists regarding the utility of home-based screening tools for amblyopia. When presenting results, I strongly encourage authors to include a brief discussion of the cost-effectiveness of amblyopia management as compared to other ophthalmic diseases (Reference: Busbee BG, Brown GC, Brown MM (2003) Cost-effectiveness of ocular interventions. *Curr Opin Ophthalmol* 14(3):132–138. <https://doi.org/10.1097/00055735-200306000-00004>)

The advice from this reviewer is noted and we will include a brief discussion of cost-effectiveness of amblyopia management in our results.

Reviewer: 2

Dr. Patricia Nelson, Texas Tech University Health Sciences Center

Comments to the Author:

Well thought out study design and well constructed appendix resources. Would be a relevant, impactful, and timely study.

Thank you for the comments.

Reviewer: 3

Dr. Anna O'Connor, University of Liverpool

Comments to the Author:

The use of home based devices for assessing children is a very important one and highly pertinent in current times. Many children have yet to be screened from reception class in 2019/2020 with more waiting from the current cohort of children. This protocol is timely and appropriate but the focus of just looking at devices to be used in a screening context is very specific. There is a potential role for devices to monitor VA at home as well, not just for detection purposes. During the pandemic there have been a number of papers evaluating the use of VA testing methods at home, some of which the authors have cited. Is there scope for evaluating the test accuracy overall rather than simply for the detection of amblyopia? To determine accuracy of any test at detecting amblyopia it would require the child to have a further assessment to diagnose amblyopia. Given the limited number of studies, it would be worthwhile considering expanding to include accuracy in general of VA tests performed at home to include measures such as test retest variability.

Thank you for the comments. It is true that there are home-based devices used to monitor vision among children, not just for detection purposes. However scoping searches carried out showed that there are only a small number of validation studies done on home-based vision monitoring tests in a paediatric cohort, which report validity measures such as correlation and test-retest variability. Furthermore, there is a systematic review already published by Samanta and colleagues on the validity of home-based vision monitoring tests among the general population. We also aim to publish a more specific systematic review summarising higher levels of evidence from diagnostic accuracy studies on the utility of these home-based test in screening for amblyogenic conditions. However, the

authors of this study are interested in the prospect of summarising evidence on the accuracy, reliability, and validity of vision monitoring methods at home for children as more validation studies are being done in this field in the future.

Two minor points - the British and Irish Orthoptic Journal is on PubMed but not Medline, is there scope for including this journal (and other relevant orthoptics journals)?

Thank you for pointing this out. We will also include relevant orthoptic journals by conducting a search on both PubMed and Medline.

Search strategy – is there a reason that the term visual acuity isn't included?

We have done a revision on the search strategy to include this as a MESH term.

VERSION 2 – REVIEW

REVIEWER	Anna O'Connor University of Liverpool
REVIEW RETURNED	10-Aug-2021
GENERAL COMMENTS	The authors have addressed the comments from the editor and reviewers and I recommend the paper be accepted for publication.